# Tight Bounds for Machine Unlearning via Differential Privacy

## Abstract

We consider the formulation of "machine unlearning" of Sekhari, Acharya, Kamath, and Suresh (NeurIPS 2021), which formalizes the so-called "right to be forgotten" by requiring that a trained model, upon request, should be able to 'unlearn' a number of points from the training data, as if they had never been included in the first place. Sekhari et al. established some positive and negative results about the number of data points that can be successfully unlearnt by a trained model without impacting the model's accuracy (the "deletion capacity"), showing that machine unlearning could be achieved by using differentially private (DP) algorithms. However, their results left open a gap between upper and lower bounds on the deletion capacity of these algorithms: our work fully closes this gap, obtaining tight bounds on the deletion capacity achievable by DP-based machine unlearning algorithms.

## 1 Introduction

Machine learning models trained on user data are now routinely used virtually everywhere, from recommendation systems to predictive models. In many cases, this user data itself includes some sensitive information (e.g., healthcare or race) or private aspects (customer habits, geographic data), sometimes even protected by law. To address this issue – that the models trained on sensitive datasets must not leak personal or private information – in a principled fashion, one of the leading frameworks is that of *differential privacy* (DP) [Dwork et al., 2006], which has *de facto* become the standard for privacy-preserving machine learning over the past decade.

At its core, DP requires that the output of a randomized algorithm $M$ not change drastically if one to modify one of the datapoints: that is, if $X, X'$ are two datasets only differing in *one* user's data, then for all possible outputs $S$ of the algorithm one should have roughly the same probability of observing $S$ under both inputs:

$$\Pr[M(X) \in S] \le e^{\varepsilon} \Pr[M(X') \in S] + \delta$$

where $\varepsilon > 0$ and $\delta \in (0, 1]$ quantify the privacy guarantee (the smaller values, the better the privacy; see Section 2 for formal definitions). Intuitively, an algorithm $M$ being $(\varepsilon, \delta)$-DP means that its output does not reveal much about any particular user's data, since the output would be nearly identical had this user's data been completely different.

While the use of differential privacy can mitigate many privacy concerns, it does come with some limitations. The first is the overhead in brings: that is, ensuring differential privacy for a learning task typically incurs an overhead in the number of data points needed to achieve the same accuracy guarantee. Perhaps more importantly, DP does not solve all possible privacy concerns: even if a ML model is trained on a sensitive dataset in a differentially private way, the dataset may still be subject to some attacks – e.g., if the server where the training data is stored is itself compromised. Somewhat

tautologically: DP is not a silver bullet, and only provides meaningful guarantees against the threat models it was meant to address.

Another type of concerns focuses on the individual *right to maintain control on one's own data*: broadly speaking, this is asking that each user can (under some reasonable circumstances) require that their personal data and information be removed from a company's collected data and trained models. This so-called "right to be forgotten," which allow people to request that their data be deleted entirely from an ML system, has been passed into legislation or is considered in some form or another by various countries or entities, prominently the European Union's General Data Protection Regulation (GDPR), the California Privacy Rights Act (CCRA), Canada's proposed Consumer Privacy Protection Act (CPPA), and most recently in Australia [Karp, 2023].

However, translating this "right to be forgotten" into practice comes with a host of challenges, starting with how to formalize it [Cohen et al., 2022] and technically implement it – which recently led to a new area of research in ML and computer science, that of *machine unlearning*. A naive technical solution would be for a given company to keep the original training set at all times, and, upon a deletion request by a user, remove this user's data from the set before retraining the whole model on the result. This, of course, comes up with two major drawbacks: first, the cost to the company, in terms of time and computational resources, of retraining a large model on a regular basis. Second, the *privacy cost*, as keeping the training set for an indefinite time in order to be able to handle the deletion requests leaves the door open to potential attacks and data breaches. Fortunately, there have been, over the past few years, a flurry of better (and more involved) approaches to machine unlearning, to handle deletion requests much more efficiently, and requiring to maintain much less of the training set (see, e.g., [Bourtoule et al., 2021, Nguyen et al., 2022], and related work below).

The above discussion, still, brings to light an important question: *is machine unlearning, paradoxically, at odds with (differential) privacy? What is the connection between the two notions: are they complementary, or is there a trade-off between them?*

This is the main question this work sets out to address. Our starting point is the formalization of machine unlearning set forth by Sekhari, Acharya, Kamath, and Suresh [Sekhari et al., 2021], itself reminiscent of the definition of DP (see Definition 2.5 for the formal statement): a pair of algorithms $(A, \bar{A})$ is an $(\varepsilon, \delta)$-*unlearning algorithm* if (1) $A \colon \mathcal{X}^* \to \mathcal{W}$ is a (randomized) learning algorithm which, given a dataset $X \subseteq \mathcal{X}^*$, outputs model parameters $A(X) \in \mathcal{W}$; and (2) $\bar{A} \colon \mathcal{X}^* \times \mathcal{W} \times \mathcal{T} \to \mathcal{W}$ which, on input a set of *deletion requests* $U \subseteq X$, previous model parameters $w$, and some succinct additional "side information" $T(X) \in \mathcal{T}$ about the original dataset, output updated model parameters $w' \in \mathcal{W}$ from which the data from $U$ has been unlearned, that is, such that

$$\Pr\big[\, \bar{A}(U, A(X), T(X)) \in W \,\big] \le e^\varepsilon \Pr\big[\, \bar{A}(\emptyset, A(X \setminus U), T(X \setminus U)) \in W \,\big] + \delta$$

and
$$\Pr\big[\, \bar{A}(\emptyset, A(X \setminus U), T(X \setminus U)) \in W \,\big] \le e^\varepsilon \Pr\big[\, \bar{A}(U, A(X), T(X)) \in W \,\big] + \delta$$

for every possible set $W \subseteq \mathcal{W}$ of model parameters. Loosely speaking, this requires that the outcomes of (a) training a model $M$ via $A$ on the dataset $X$ then unlearning some of the original training data $U \subseteq X$ from $M$ using $\bar{A}$, and (b) training a model $M'$ via $A$ directly on the dataset $X \setminus U$ then unlearning nothing via $\bar{A}$, be nearly indistinguishable.

In their paper, Sekhari et al. [Sekhari et al., 2021] focus on genralization guarantees of unlearning algorithm, i.e., what can be achieved by unlearning algorithms when focusing on population loss, namely, when aiming to minimize

$$F(w) \coloneqq \mathbb{E}_{x \sim \mathcal{D}}[f(w, x)]$$

given a prespecified loss function $f \colon \mathcal{W} \times \mathcal{X} \to \mathbb{R}$, where the expectation is over the draw of a new datapoint from the underlying distribution $p$ on the sample space. The quality of a learning algorithm $A$ is then measured by the expected excess risk

$$R(f, A) \coloneqq \mathbb{E}\left[ F(A(X)) - \inf_{w^* \in \mathcal{W}} F(w^*) \right]$$

where the expectation is taking over the random choice of a dataset $X \sim \mathcal{D}^n$ of size $n$, and the randomness of $A$ itself. The focus of [Sekhari et al., 2021], as is ours, is then to quantify the *deletion capacity* achievable for $(\varepsilon, \delta)$-unlearning given a prespecified loss function, that is, the maximum

number of data points one can ask to be forgotten (maximum size of the subset $U$) before the excess risk increases by more than some threshold (see Definition 2.6).

In their paper, [Sekhari et al., 2021] draw a connection between DP learning algorithms and unlearning ones, showing that DP learning algorithms do provide *some* unlearning guarantees out-of-the-box, and that one can achieve non-trivial unlearning guarantees for convex loss functions by leveraging the literature on differentially private optimization and learning. One of their main results is showing that these DP-based unlearning algorithms, which crucially *do not rely on any side information* (the additional input $T(X) \in \mathcal{T}$ provided to the unlearning algorithm $\bar{A}$) can handle strictly fewer deletion requests than general unlearning algorithms which *do* rely on such side information.

Their results, however, do not fully characterize the deletion capacity of these "DP-based" machine unlearning algorithms, leaving a significant gap between their upper and lower bounds. We argue that fully understanding this quantity is crucial, as DP-based unlearning algorithms are *exactly* those for which there is no conflict between the two notions of DP and unlearning – *instead, this class of algorithms is the one for which they work hand in hand.* This is in contrast to the more general unlearning algorithms relying on maintaining and storing side information about the training set, as this side information can make their deployment susceptible to privacy breaches.

## 1.1 Our contributions

The main contribution of our paper is a tight bound on the "amount of unlearning" achievable by *any* machine unlearning algorithm which does not rely on side information. For the sake of exposition, we state in this section informal versions of our results.

**Theorem 1.1** (Unlearning For Convex Loss Functions (Informal; see Theorems 3.1 and 3.3)). *Let $f \colon \mathcal{W} \times \mathcal{X} \to \mathbb{R}$ be a 1-Lipschitz convex loss function, where $\mathcal{W} \subseteq \mathbb{R}^d$ is the parameter space. There exists an $(\varepsilon, \delta)$-machine unlearning algorithm which, trained on a dataset $S \subseteq \mathcal{X}^n$, does not store any side information about the training set besides the learned model, and can unlearn up to*

$$m = O\left(\frac{n\varepsilon\alpha}{\sqrt{d\log(1/\delta)}}\right)$$

*datapoints without incurring excess population risk greater than $\alpha$. Moreover, this is tight: there exists a 1-Lipschitz linear loss function such that no machine unlearning algorithm can unlearn $\Omega(\frac{n\varepsilon\alpha}{\sqrt{d\log(1/\delta)}})$ data points without excess population risk $\alpha$, unless it stores side information.*

Our techniques also allow us to easily establish the analogue for *strongly* convex optimization:

**Theorem 1.2** (Unlearning For Strongly Convex Loss Functions (Informal)). *Let $f \colon \mathcal{W} \times \mathcal{X} \to \mathbb{R}$ be a 1-Lipschitz strongly convex loss function. There exists an $(\varepsilon, \delta)$-machine unlearning algorithm which, trained on a dataset $S \subseteq \mathcal{X}^n$, does not store any side information about the training set besides the learned model, and can unlearn up to*

$$m = O\left(\frac{n^2\varepsilon\alpha}{d\log(1/\delta)}\right)$$

*datapoints without incurring excess population risk greater than $\alpha$. Moreover, this is tight.*

We note that, prior to our work, only bounds for the convex loss function case were known, with an upper bound of $m = \tilde{O}(n\varepsilon\alpha/\sqrt{d\log(e^\varepsilon/\delta)})$ (loose by polylogarithmic factors for $\varepsilon = O(1)$, as well as an $1/\sqrt{\varepsilon}$ factor for $\varepsilon \gg 1$) and a limited lower bound stating that $m \geq 1$ is only possible if $n\varepsilon/\sqrt{d} = \Omega(1)$.

Our next contribution, motivated by the similarity of the formalisations of machine unlearning (without side information) and that of differential privacy, is to establish the analogue of key properties of DP for machine unlearning, namely, *post-processing* and *composition* of machine unlearning algorithms. To do so, we first identify a natural property of machine unlearning algorithms, which, when satisfied, will allow for the composition properties:

**Assumption 1.3** (Unlearning Laziness). *An unlearning algorithm $(\bar{A}, A)$ is said to be lazy if, when provided with an empty set of deletion requests, the unlearning algorithm $\bar{A}$ does not update the model. That is, $\bar{A}(\emptyset, A(X), T(X)) = A(X)$ for all datasets $X$.*

128   We again emphasize that this laziness property is not only intuitive, it is also satisfied by several
129   existing unlearning algorithms, and in particular those proposed in Sekhari et al. [2021].

130   **Theorem 1.4** (Post-processing of unlearning). *Let $(\bar{A}, A)$ be an $(\varepsilon, \delta)$-unlearning algorithm taking*
131   *no side information. Let $f : \mathcal{W} \to \mathcal{W}$ be an arbitrary (possibly randomized) function. Then $(f \circ \bar{A}, A)$*
132   *is also an $(\varepsilon, \delta)$-unlearning algorithm.*

133   Under our laziness assumption, we also establish the following:

134   **Theorem 1.5** (Chaining of unlearning). *Let $(\bar{A}, A)$ be a lazy $(\varepsilon, \delta)$-unlearning algorithm taking*
135   *no side information, and able to handle up to $m$ deletion requests. Then, the algorithm which uses*
136   *$(\bar{A}, A)$ to sequentially unlearn $k$ disjoint deletion requests $U_1, \ldots, U_k \subseteq X$ such that $|\cup_i U_i| \leq m$,*
137   *outputting*

$$\bar{A}(U_k, \ldots, \bar{A}(U_1, A(X)) \ldots)$$

138   *is an $(\varepsilon', \delta')$-unlearning algorithm, with $\varepsilon' = k\varepsilon$ and $\delta' = \delta \cdot \frac{e^{k\varepsilon}-1}{e^{\varepsilon}-1} = O(k\delta)$ (for $k = O(1/\varepsilon)$).*

139   and, finally,

140   **Theorem 1.6** (Advanced composition of unlearning). *Let $(\bar{A}_1, A), \ldots, (\bar{A}_k, A)$ be lazy $(\varepsilon, \delta)$-*
141   *unlearning (with common learning algorithm $A$) taking no side information, and define the composi-*
142   *tion of those unlearning algorithms, $\tilde{A}$ as*

$$\tilde{A}(U, A(X)) = f\big(\bar{A}_1(U, A(X)), \ldots, \bar{A}_k(U, A(X))\big).$$

143   *where $f : \mathcal{W}^k \to \mathcal{W}$ is any (possibly randomized) function. Then, for every $\delta' > 0$, $(\tilde{A}, A)$ is an*
144   *$(\varepsilon', \delta')$-unlearning taking no side information, where $\varepsilon' = \frac{k}{2}\varepsilon^2 + \varepsilon\sqrt{2k \ln (1/\delta')}$.*

## 1.2   Related work

146   Albeit recent, the field of machine unlearning has already received considerable attention from the ML
147   community, with an array of studies and papers focusing on practical solutions and their empirical
148   performance. We focus in this section on the works most closely related to ours, mostly theoretical.
149   As discussed earlier, the goal of machine unlearning (Bourtoule et al. [2021]) is to delete what models
150   have learned from data. This problem coincides tangentially with the idea of differential privacy as
151   they both requires to minimize the effect of a (or a group of) sample. The original, stringent definition
152   of unlearning requires $\varepsilon = 0$ (full deletion of the user's data, as if it had never been included in the
153   training set in the first place) in contrast to differential privacy that allows $\varepsilon > 0$, leaving a possibility
154   for "memorization." To relax this definition, Ginart et al. [2019] proposed the probabilistic version of
155   unlearning.

156   Prior theoretical work of unlearning are mostly disjoint from the differential privacy literature,
157   in spite of a general recognition that the two notions aim to address related issues. Most works
158   on machine unlearning mainly focus on empirical risk minimization of approximate unlearning
159   algorithms (Guo et al. [2020], Izzo et al. [2020]), which seeks to find an approximate minimizer on
160   the remaining dataset after deletion. Closest to our work is the recent paper of Sekhari et al. [2021],
161   which formulated the notion of machine unlearning used in our paper and focused on population
162   loss minimization of approximating unlearning algorithm (i.e., allowing $\varepsilon > 0$). Their objectives,
163   however, were somewhat orthogonal to ours, as they focused for a large part on minimizing the space
164   requirements for the side information $T(X)$ provided to the unlearning algorithm (while our paper
165   focuses on algorithms which do *not* rely on any such side information, prone to potential privacy
166   leaks). While their work, to motivate this focus, established partial bounds on the deletion capacity
167   of unlearning algorithm that do not take in extra statistics, these bounds were not tight, and one
168   of our main contributions is closing this gap. Following Sekhari et al. [2021], the notion of *online*
169   unlearning algorithm – which receive the deletion requests sequentially – was put forward and studied
170   in Suriyakumar and Wilson [2022], again with memory efficiency with respect to the side information
171   in mind; however, their primary focus is on the empirical performance of unlearning algorithm.

172   Another work closely to ours is the notion of *certified data removal* proposed by Guo et al. [2020].
173   The main difference between $(\varepsilon, \delta)$-certified removal and the definition from Sekhari et al. [2021] is
174   that, in the former, the unlearning mechanism requires access not only to the samples to be deleted
175   (the set $U \subseteq X$), but also to the full original training set $X$: this is exactly the type of constraints our
176   work seeks to avoid, due to the risk of data breach this entails.

 **1.3 Organization of the paper**

 We first provide the necessary background and notion on differential privacy, learning, and the
 formulation of machine unlearning used throughout the paper in Section 2. We then provide a detailed
 outline of the proof of our main result, Theorem 1.1, in Section 3, before concluding with a discussion
 of results and future work in Section 4.

 Due to space constraints, the details of all other results, as well as omitted proofs, are deferred to the
 Supplemental.

# 2 Preliminaries

In this section, we recall some notions and results we will extensively rely on in our proofs and
theorems, starting with differential privacy.

## 2.1 Differential Privacy

**Definition 2.1** ((Central) Differential Privacy (DP)). *Fix $\varepsilon > 0$ and $\delta \in [0, 1]$. An algorithm
$M \colon \mathcal{X}^n \to \mathcal{Y}$ satisfies $(\varepsilon, \delta)$-differential privacy (DP) if for every pair of neighboring datasets $X, X'$,
and every (measurable) subset $S \subseteq \mathcal{Y}$:*

$$\Pr[\, M(X) \in S \,] \le e^\varepsilon \Pr[\, M(X') \in S \,] + \delta.$$

*We further say that $M$ satisfies* pure *differential privacy ($\varepsilon$-DP) if $\delta = 0$, otherwise it is* approximate
*differential privacy.*

We now recall another notion of differential privacy in terms of Renyi Divergence, from Bun and
Steinke [2016].

**Definition 2.2** (Zero-Concentrated Differential Privacy (zCDP)). *A randomized algorithm $M : \mathcal{X}^n \to \mathcal{Y}$ satisfies $(\xi, \rho)$-zCDP if for every neighboring datasets (differing on a single entry) $X, X' \in \mathcal{X}^n$,
and $\forall \alpha \in (1, \infty)$:*
$$\mathrm{D}_\alpha(M(X)\|M(X')) \le \xi + \rho\alpha$$
*where $\mathrm{D}$ is the $\alpha$-Renyi divergence between distributions of $M(X)$ and $M(X')$. We say that $M$ is
$\rho$-zCDP when $\xi = 0$.*

We use the following group privacy property of zCDP in the proof later.

**Proposition 2.3** ($k$-distance group privacy of $\rho$-zCDP [Bun and Steinke, 2016, Proposition 1.9]). *Let
$M : \mathcal{X}^n \to \mathcal{Y}$ satisfy $\rho$-zCDP. Then, $M$ is $(k^2\rho)$-zCDP for every $X, X' \in \mathcal{X}^n$ that differs in at most
$k$ entries.*

## 2.2 Learning

We also will require some definitions on learning, specifically with respect to minimizing population
loss. Fix any loss function $f \colon \mathcal{W} \times \mathcal{X}$, where $\mathcal{W}$ is the (model) parameter space and $\mathcal{X}$ is the sample
space. Then, the generalization loss is defined as

$$F(w) \coloneqq \mathbb{E}_{x \sim p}[f(w, x)]$$

in which the expectation is over the distribution of $x$ (one sample) and $w$ is the learning output. Let
$F^* = \min_{w \in \mathcal{W}} F(w)$ be the minimizer of population risk and $w^*$ is the corresponding minimizer.

Define learning algorithm $A : \mathcal{X}^n \to \mathcal{W}$ that takes in dataset $S \in \mathcal{X}^n$ and returns hypothesis
$w \coloneqq A(S) \in \mathcal{W}$. The excess risk is given by:

$$\mathbb{E}[F(A(S))] - F^*$$

where the expectation is over the randomness of $A$ and $S$.

Hence, we could define the sample complexity as following ([Sekhari et al., 2021, Definition 1]),
which is analogous to deletion capacity, in which will be stated later.

**Definition 2.4** (Sample complexity of learning). *The $\alpha$-sample complexity of a problem is defined as:*

$$n(\alpha) \coloneqq \min\{n \mid \exists A \text{ s.t. } \mathbb{E}[F(A(S))] - F^* \le \alpha, \ \forall \mathcal{D}\}$$

### 2.3 Unlearning

As previously discussed, we rely on the definition of unlearning proposed in by Sekhari et al. [2021], and maintain same notation. Note that $T(S)$ denotes the data statistics (which could be the entire dataset $S$ or any form of statistic) available to $\bar{A}$.

**Definition 2.5** (($\varepsilon, \delta$)-unlearning). *For all $S$ of size $n$ and delete requests $U \subseteq S$ such that $|U| \leq m$, and $W \subseteq \mathcal{W}$, a learning algorithm $A$ and an unlearning algorithm $\bar{A}$ is ($\varepsilon, \delta$)-unlearning if:*

$$\Pr\big[\,\bar{A}(U, A(S), T(S)) \in W\,\big] \leq e^{\varepsilon} \Pr\big[\,\bar{A}(\emptyset, A(S \setminus U), T(S \setminus U)) \in W\,\big] + \delta$$

*and*

$$\Pr\big[\,\bar{A}(\emptyset, A(S \setminus U), T(S \setminus U)) \in W\,\big] \leq e^{\varepsilon} \Pr\big[\,\bar{A}(U, A(S), T(S)) \in W\,\big] + \delta,$$

Our results will be phrased in terms of the deletion capacity, which captures the number of deletion requests an unlearning algorithm can handle before seeing a noticeable drop in its output's accuracy:

**Definition 2.6** (Deletion Capacity). *Let $\varepsilon, \delta > 0$, $S$ be a dataset of size $n$ drawn i.i.d. from $\mathcal{D}$ and let $\ell(w, z)$ be a loss function. For a pair of learning and unlearning algorithm $A, \bar{A}$ that are ($\varepsilon, \delta$)-unlearning, the deletion capacity $m_{\varepsilon, \delta}^{A, \bar{A}}$ is defined as the maximum size of deletions requests set $|U|$ that we can unlearn without doing worse in excess population risk than $\alpha$:*

$$m_{\varepsilon, \delta}^{A, \bar{A}}(\alpha) := \max\{m \mid \mathbb{E}\Big[\max_{U \subseteq S: |U| \leq m} F(\bar{A}(U, A(S), T(S))) - F^*\Big] \leq \alpha\}$$

*where $F^* := \min_{A(S) \in \mathcal{B}} F(\bar{A}(U, A(S), T(S)))$.*

## 3 Main result

In this section, we provide a detailed outline of our main result on unlearning for convex loss functions, Theorem 1.1, for which we prove the upper and lower bounds separately.

**Theorem 3.1** (Deletion capacity from unlearning via DP, Lower Bound). *Suppose $\mathcal{W} \subseteq \mathbb{R}^d$, and fix any Lipschitz convex loss function. Then there exists a lazy ($\varepsilon, \delta$)-unlearning algorithm ($\bar{A}, A$), where $\bar{A}$ has the form $\bar{A}(U, A(S), T(S)) := A(S)$ (and thus, in particular, takes no side information) with deletion capacity*

$$m_{\varepsilon, \delta}^{A, \bar{A}}(\alpha) \geq \Omega\left(\frac{\varepsilon n \alpha}{\sqrt{d \log(1/\delta)}}\right)$$

*where the constant in the $\Omega(\cdot)$ only depends on the properties of the loss function.*

*Proof.* Our proof follows the same general outline as that of Sekhari et al. [2021], with a key difference which allows us to derive the tight bound. Namely, we start, similarly, from the observation that any ($\varepsilon, \delta$)-DP learning algorithm $A$ whose privacy guarantee is with respect to edit distance $m$ between datasets (instead of the usual neighboring relation) readily implies an ($\varepsilon, \delta$)-machine unlearning algorithm ($\bar{A}, A$), where $\bar{A}(w) = w$ (i.e., the "unlearning part" returns its input unchanged).

However, we depart from previous work by how we obtain this ($\varepsilon, \delta$)-DP algorithm $A$ with respect to edit distance $m$. The key insight is that instead of starting with any good approximate DP learning algorithm and using the grouposition property of DP to "upgrade" it to $m$-edit distance, we instead start with a good *zCDP* learning algorithm. Indeed, zCDP has much tigher grouposition properties than approximate DP (cf. Proposition 2.3), which in turn leads to better parameters when applying grouposition to achieve DP to groups up to size $m$: specifically, starting with a $\rho^2/2$-zCDP standard privacy guarantee (for groups of size 1) we would by Proposition 2.3 obtain $(m^2 \rho^2/2)$-zCDP for neighboring datasets differin in up to $m$ entries. Leveraging then the standard conversion from concentrated to approximate DP [Bun and Steinke, 2016], this implies, for every $\delta > 0$, an ($\varepsilon, \delta$)-DP guarantee for groups of size $m$, where $\varepsilon = O(m\rho\sqrt{\log(1/\delta)})$. Thus, choosing $\rho = \Theta\left(\frac{\varepsilon}{m\sqrt{\ln(1/\delta)}}\right)$ would suffice to achieve the desired end privacy guarantee on $A$ (with respect to edit distance up to $m$), and thus the ($\varepsilon, \delta$)-unlearning one for ($\bar{A}, A$).

To do so, however, we crucially need to start with a sufficiently good private learning algorithm $A$ with zCDP guarantees, instead of approximate DP. Fortunately for us, such an algorithm is provided by [Feldman et al., 2020, Theorem 1]:

**Lemma 3.2** (zCDP mini-batch noisy SGD Feldman et al. [2020])*. Fix any $L$-Lipschitz convex loss function over a convex subset $\mathcal{B}$ of $\mathbb{R}^d$ of diameter $D$. Then there exists an algorithm $A$ which satisfies $(\rho^2/2)$-zCDP with excess population loss:*

$$\mathbb{E}\left[F(\theta) - \min_{\theta \in \mathcal{B}} F(\theta)\right] \leq O\left(DL \cdot \left(\frac{1}{\sqrt{n}} + \frac{\sqrt{d}}{\rho n}\right)\right) \tag{1}$$

*where the expectation is taken over the randomness of $A$.*

By the above discussion, using this zCDP-private learning algorithm with $\rho = \Theta\left(\frac{\varepsilon}{m\sqrt{\ln(1/\delta)}}\right)$, we get an excess population loss bounded by

$$O\left(DL\left(\frac{1}{\sqrt{n}} + \frac{m\sqrt{d\ln(1/\delta)}}{\varepsilon n}\right)\right) \tag{2}$$

It only remains to show how the claimed deletion capacity bound frollows from this excess population risk guarantee. Construct, as discussed earlier, an unlearning algorithm $\bar{A}$ that returns the input without making any changes (and in particular does not require any additional statistics $T(S)$, and satisfies the laziness assumption). Since $A$ is $(\varepsilon, \delta)$-DP, for any set $U \subseteq S, |U| = m$, and $W \subseteq \mathcal{W}$,

$$\Pr[\, A(S) \in W \,] \leq e^\varepsilon \Pr[\, A(S') \in W \,] + \delta$$
$$\Pr[\, A(S') \in W \,] \leq e^\varepsilon \Pr[\, A(S) \in W \,] + \delta$$

. But since $\bar{A}(U, A(S)) = A(S)$, this readily yields, letting $S' := S \setminus U$:

$$\Pr[\, \bar{A}(U, A(S)) \in W \,] \leq e^\varepsilon \Pr[\, \bar{A}(\emptyset, A(S')) \in W \,] + \delta$$
$$\Pr[\, \bar{A}(\emptyset, A(S')) \in W \,] \leq e^\varepsilon \Pr[\, \bar{A}(U, A(S)) \in W \,] + \delta$$

which implies that $(A, \bar{A})$ is indeed $(\varepsilon, \delta)$-unlearning for $U$ of size (up to) $m$.

Recalling the definition of deletion capacity (Definition 2.6), we finally deduce from (2) the deletion capacity with excess population risk less than $\alpha$:

$$m_{\varepsilon,\delta}^{A,\bar{A}}(\alpha) \geq m = \Omega\left(\frac{\varepsilon n \alpha}{\sqrt{d\ln(1/\delta)}}\right)$$

where the $O(\cdot)$ hides constant factors depending only on the loss function (namely, the Lipschitz function $L$, and the diameter $D$). $\qquad\square$

**Theorem 3.3** (Deletion capacity from unlearning via DP, Upper Bound)*. There exists a Lipschitz convex loss function (indeed,* linear*) for which any $\varepsilon, \delta)$-unlearning algorithm $(\bar{A}, A)$ which takes no side information must have deletion capacity*

$$m_{\varepsilon,\delta}^{A,\bar{A}}(\alpha) \leq O\left(\frac{\varepsilon n \alpha}{\sqrt{d\log(1/\delta)}}\right).$$

*Detailed Proof Sketch.* We will consider the following linear (and therefore convex and Lipschitz) loss function $\mathcal{L}(\theta, S)$:

$$\mathcal{L}(\theta, S) := -\langle\theta, \sum_{i=1}^{n} x_i\rangle \tag{3}$$

for dataset $S$ of $n$ points $x_1, \ldots, x_n \in \{-\frac{1}{\sqrt{d}}, \frac{1}{\sqrt{d}}\}^d$. We also define the 1-way marginal query, i.e. average, as:

$$q(S) := \frac{1}{n}\sum_{i=1}^{n} x_i. \tag{4}$$

To establish our deletion capacity lower bound with respect to this loss function, we will proceed in three stages: the first, relatively standard, is to relate population loss (what we are interested in) to *empirical* loss – which allows us to focus on the existence of a "hard dataset." The second step

is then to establish a sample complexity lower bound on the empirical risk (for this loss function) of any $(\varepsilon, \delta)$-DP algorithm, via a reduction to differentially private computing of 1-marginals. This step is similar to the one underlying the (weaker) lower bound of Sekhari et al. [2021] (itself relying on an argument of [Bassily et al., 2019]), although a more careful choice of building blocks for the reduction already allows us to obtain an improvement by logarithmic factors.

Third, lift this DP lower bound to a stronger lower bound for DP with respect to edit distance $m$. This step is quite novel, as it morally corresponds to establishing the converse of the grouposition property of differential privacy (for our specific setting), a converse which does *not* hold in general. Our argument, relatively simple, will crucially rely on the linearity of our loss function.

We omit the details of the first step (reduction from population to empirical loss) in this detailed outline, as it is quite standard. For the second step, our starting point is the following lower bound of Steinke and Ullman:

**Theorem 3.4** (Lower bound for one-way marginals [Steinke and Ullman, 2016, Main Theorem]). *For every $\varepsilon \in (0,1)$, every function $\delta = \delta(n)$ such that $\delta \geq 2^{-o(n)}$ and $\delta \leq 1/n^{1+\Omega(1)}$), and for every $\alpha \leq 1/10$, if $A : \{\pm 1\}^{n \times d} \to [\pm 1]^d$ is $(\varepsilon, \delta)$-differentially private and $\mathbb{E}[\|\mathcal{A}(S) - q(S)\|_1] \leq \alpha d$ (i.e., with average-case accuracy $\alpha$) for all $S \in \{\pm 1\}^{n \times d}$, then we must have*

$$n \geq \Omega\left(\frac{\sqrt{d \ln(1/\delta)}}{\varepsilon \alpha}\right).$$

Using this lower bound as a blackbox, we then can adapt the argument of [Bassily et al., 2014, Lemma 5.1, Part 2] to obtain the following stronger result:

**Lemma 3.5** (Lower bound for Privately Computing 1-way Marginals). *Let $n, d \in \mathbb{N}, \varepsilon > 0, 2^{-on} \leq \delta(n) \leq 1/n^{1+\Omega(1)}$. For all $\alpha \leq 1/10$, if $\mathcal{A}$ is $(\varepsilon, \delta)$-differentially private and $S \subseteq \{\pm \frac{1}{\sqrt{d}}\}^{n \times d}$:*

$$\mathbb{E}[\|\mathcal{A}(S) - q(S)\|_2] = \min\left(\alpha, \Omega\left(\frac{\sqrt{d \ln(1/\delta)}}{n\varepsilon}\right)\right),$$

*where $q(S) = \frac{1}{n}\sum_{i=1}^{n} x_i$ as before.*

Combining the above with the argument strategy of [Bassily et al., 2014, Theorem 5.3] finally yields the main lemma for the second step of our proof:

**Lemma 3.6** (Lower bound on empirical loss of $(\varepsilon, \delta)$-DP algorithms). *Let $n, d \in \mathbb{N}, \varepsilon > 0$, and $\delta = o(1/n)$. For every $(\varepsilon, \delta)$-differentially private algorithm with output $\theta^{priv}$, there is a dataset $S = \{x_1, \ldots, x_n\} \subseteq \{-\frac{1}{\sqrt{d}}, \frac{1}{\sqrt{d}}\}^d$ such that*

$$\mathbb{E}\left[\mathcal{L}(\theta^{priv}, S) - \mathcal{L}(\theta^*, S)\right] = \min\left(\alpha^2, \Omega\left(\frac{d \log(1/\delta)}{n^2 \varepsilon^2}\right)\right)$$

*where $\theta^* := \frac{\sum_{i=1}^{n} x_i}{\|\sum_{i=1}^{n} x_i\|_2}$ is the minimizer of $\mathcal{L}(\theta, S) := -\langle \theta, \sum_{i=1}^{n} x_i \rangle$.*

The above lemma establishes a lower bound on the empirical loss of any $(\varepsilon, \delta)$-differentially private algorithm. To derive from this our claimed lower bound on unlearning algorithms, we need to introduce a dependence on $m$, the deletion capacity (i.e., number of points to unlearn). This is done in the last (third) step of our argument, via a reduction which establishes a (restricted) converse to the grouposition property of DP.

Recall that an algorithm $M : \mathcal{X}^n \to \mathcal{Y}$ satisfies $(\varepsilon, \delta)$-DP for edit distance $m$ if for every pair of neighboring datasets $X, X'$ that differ in up to $m$ entries, and every $S \subseteq \mathcal{Y}$:

$$\Pr[M(X) \in S] \leq e^\varepsilon \Pr[M(X') \in S] + \delta.$$

We apply this $m$-edit distance $(\varepsilon, \delta)$-DP on Lemma 3.6 by a reduction that shows: for any differentially private algorithm with respect to edit distance at most $m$ must incur an empirical loss given by Lemma 3.6.

**Lemma 3.7.** *Suppose there exists an $m$-edit distance $(\varepsilon, \delta)$-DP algorithm $\mathcal{M}$ that takes in a dataset $S$ of size $n$ to approximate $q(S)$ (as defined in (4)), with empirical loss $\gamma$. Then, we can construct a 1-edit distance (i.e., standard) $(\varepsilon, \delta)$-DP algorithm $\mathcal{M}'$ that, on input a dataset $S'$ of $N = n/m$ data points, approximates $q(S')$ to error $\gamma$.*

*Proof.* The reduction is quite simple: given $\mathcal{M}$, construct $\mathcal{M}'$ as follows for $N = \frac{n}{m}$ inputs:

$$\mathcal{M}'(x_1, \ldots, x_N) = \mathcal{M}(\underbrace{x_1, \ldots, x_1}_{m}, \underbrace{x_2, \ldots, x_2}_{m}, \ldots, \underbrace{x_N, \ldots, x_N}_{m}).$$

We immediately have that $\mathcal{M}'$ is $(\varepsilon, \delta)$-DP for the usual 1-edit distance between datasets, since $\mathcal{M}$ is DP with respect to edit distance $m$. The sample complexity and error bound then follows direction from $n = N \times m$, where $n \geq N, N \in \mathbb{N}, m \geq 1$, and the fact that $q(x_1, \ldots, x_N) = q(x_1, \ldots, x_1, x_2, \ldots, x_2, \ldots, x_N, \ldots, x_N)$ due to the definition of $q$. $\qquad\square$

Combining Lemma 3.7 with Lemma 3.6, we get that any $m$-edit distance $(\varepsilon, \delta)$-DP algorithm $\mathcal{M}$ approximating $q$ on datasets of size $n = N \times m$ must have error $\gamma$ at least

$$\min\left(\alpha^2, \Omega\left(\frac{d \log(1/\delta)}{N^2 \varepsilon^2}\right)\right) = \min\left(\alpha^2, \Omega\left(\frac{m^2 d \log(1/\delta)}{n^2 \varepsilon^2}\right)\right)$$

which, reorganising the terms and recalling the definition of deletion capacity, yields the claimed bound on $m_{\varepsilon, \delta}^{A, \bar{A}}$. $\qquad\square$

We note that the proof of Theorem 1.2 follows from a very similar argument; we refer the reader to the Supplemental for details.

## 4   Discussion and future work

Our work fully characterized deletion capacity of any unlearning algorithm $(\bar{A}, A)$ minimizing population risk under both convex and strongly convex loss functions, when only given the model parameters (output of the learning algorithm) and the set of deletion requests. This restriction, namely that the unlearning algorithm does not rely on any additional side information, is motivated by the potential privacy risks storing (non-private) side information can entail.

We hope our work will lead to further study of the interplay between differential privacy and machine unlearning, and to additional study of "DP-like" properties of machine unlearning, such as the postprocessing and composition properties our present work identified. In view of the myriad applications these properties have had in privacy-preserving algorithm design, we believe that their analogue for machine unlearning will prove very useful.

We leave for future work the question of which unlearning guarantees can be obtained from *pure* differentially private algorithms, and of whether variants of the standard threat model for differential privacy (specifically, pan-privacy, or privacy under continual observation) could have implications for machine unlearning in an online setting where deletion requests come sequentially.

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
