# Tight Bounds for Machine Unlearning via Differential Privacy (Supplementary)

## 1 Proof of Theorem 3.1 (Lower Bound)

**Theorem 1.1** (Deletion capacity from unlearning via DP, Lower Bound (Theorem 3.1 in Submission))**.**
*Suppose $\mathcal{W} \subseteq \mathbb{R}^d$, and fix any Lipschitz convex loss function. Then there exists a lazy $(\varepsilon, \delta)$-unlearning
algorithm $(\bar{A}, A)$, where $\bar{A}$ has the form $\bar{A}(U, A(S), T(S)) := A(S)$ (and thus, in particular, takes
no side information) with deletion capacity*

$$m_{\varepsilon,\delta}^{A,\bar{A}}(\alpha) \geq \Omega\left(\frac{\varepsilon n \alpha}{\sqrt{d \log{(1/\delta)}}}\right)$$

*where the constant in the $\Omega(\cdot)$ only depends on the properties of the loss function.*

We first restate some useful results before diving into the proof, starting with some results on
Concentrated DP (zCDP).

**Proposition 1.2** ($k$-distance group privacy of $\rho$-zCDP [Bun and Steinke, 2016, Proposition 1.9])**.** *Let
$M : \mathcal{X}^n \to \mathcal{Y}$ satisfy $\rho$-zCDP. Then, $M$ is $(k^2 \rho)$-zCDP for every $X, X' \in \mathcal{X}^n$ that differs in at most
$k$ entries.*

**Lemma 1.3** (zCDP mini-batch noisy SGD Feldman et al. [2020])**.** *Fix any $L$-Lipschitz convex loss
function over a convex subset $\mathcal{B}$ of $\mathbb{R}^d$ of diameter $D$. Then there exists an algorithm $A$ which satisfies
$(\rho^2/2)$-zCDP with excess population loss:*

$$\mathbb{E}\left[F(\theta) - \min_{\theta \in \mathcal{B}} F(\theta)\right] \leq O\left(DL \cdot \left(\frac{1}{\sqrt{n}} + \frac{\sqrt{d}}{\rho n}\right)\right)$$

*where the expectation is taken over the randomness of $A$.*

*Proof of Theorem 1.1.* The proof follows the same setting as in Sekhari et al. [2021]. The main
change is that we apply group privacy bounds in terms of zCDP instead of the standard DP guarantee
provided by [Bassily et al., 2019, Theorem 3.2].

We first establish a tighter bound for algorithm that achieves $m$-entries group privacy via Lemma 1.3.
Feldman et al. [2020] provides a zCDP version of [Bassily et al., 2019, Theorem 3.2] with $\rho^2/2$-zCDP,
hence by group privacy, we yield $\frac{m^2 \rho^2}{2}$-zCDP by Proposition 1.2 for neighboring datasets differing
in $m$ entries. Then, translating $\frac{m^2 \rho^2}{2}$-zCDP to $(\varepsilon, \delta)$-DP yields $\varepsilon = O\left(m\rho\sqrt{\log{(1/\delta)}}\right)$.

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

68 *$[\frac{M-1}{n}, \frac{M+1}{n}]$, where $M = \Omega(\min(n\alpha, \frac{\sqrt{d \ln(1/\delta)}}{\varepsilon}))$.*

69 *Proof of Lemma 2.3.* Our proof follows the same outline as in Bassily et al. [2014], but using the
70 result of Theorem 2.2 as a black-box instead of the packing argument of Bassily et al. [2014]. Before
71 doing so, we have to translate the result from Theorem 2.2 into our setting, and handle the slightly
72 different choice of parameterization ($\{\pm 1\}^d$ instead of $\{\pm 1/\sqrt{d}\}^d$).

73 Let $n_\alpha := C \cdot \frac{\sqrt{d \ln(1/\delta)}}{\varepsilon \alpha}$, where $C > 0$ is (strictly smaller than) the constant hidden in the $\Omega(\cdot)$
74 of Theorem 2.2. By contradiction, suppose that, for some $n \leq n_\alpha$, we have an $(\varepsilon, \delta)$-differentially
75 private algorithm $\mathcal{A}$ that takes in a dataset $S \subseteq \{\pm \frac{1}{\sqrt{d}}\}^{n \times d}$ and outputs an estimate $\mathcal{A}(S)$ of $q(S)$
76 with expected $L_2$ error $\alpha$. Rescaling, we get that the algorithm $\mathcal{A}'$ which, on input $S' \subseteq \{\pm 1\}^{n \times d}$,
77 computes $S := S'/\sqrt{d} \subseteq \{\pm \frac{1}{\sqrt{d}}\}^{n \times d}$ and outputs $\sqrt{d} \cdot \mathcal{A}(S)$ is (1) $(\varepsilon, \delta)$-DP by post-processing,
78 and (2) since $q$ is linear, has error related to that of $\mathcal{A}$ by

$$\mathbb{E}[\|\mathcal{A}'(S') - q(S')\|_2] = \sqrt{d} \cdot \mathbb{E}[\|\mathcal{A}(S) - q(S)\|_2] \leq \sqrt{d} \cdot \alpha \qquad (4)$$

79 However, by Theorem 2.2, $\mathcal{A}'$ must have expected $L_1$ error at least $\alpha d$ since $n \leq n_\alpha$. By Cauchy–
80 Schwarz,

$$\alpha d < \mathbb{E}[\|\mathcal{A}'(S') - q(S')\|_1] \overset{\text{CS}}{\leq} \sqrt{d} \cdot \mathbb{E}[\|\mathcal{A}'(S') - q(S')\|_2] \overset{(4)}{\leq} \sqrt{d} \cdot (\alpha \sqrt{d}) = \alpha d$$

81 leading to a contradiction. So for $n \leq n_\alpha$, any $(\varepsilon, \delta)$-DP algorithm to estimate $q$ must have expected
82 $L_2$ error at least $\alpha$, i.e., $\mathbb{E}[\|\mathcal{A}(S) - q(S)\|_2] \geq \alpha$. Further, one can see by inspection of the proof
83 of Theorem 2.2 that $\|q(S)\|_2$ satisfies the assumption in the "Moreover."

84 Now, for $n \geq n_\alpha$ (assume, for simplicity and without loss of generality, that $n - n_\alpha$ is even), we use
85 a padding argument to establish the other part of the bound. Let $\mathcal{A}$ be any $(\varepsilon, \delta)$-differentially private
86 algorithm for answering $q$ on datasets of size $n$. Suppose for the sake of contradiction, that $\mathcal{A}$ satisfies

$$\mathbb{E}[\|\mathcal{A}(S) - q(S)\|_2] < \frac{n_\alpha}{n} \cdot \alpha \qquad (5)$$

87 for every dataset $S$ of size $n$.

88 Fix an arbitrary point $\mathbf{c} \in \{\pm 1/\sqrt{d}\}^d$. Given any dataset $S = (x^{(1)}, \ldots, x^{(n_\alpha)}) \in \{\pm 1\}^{d \times n_\alpha}$ of
89 size $n_\alpha$, we construct $\hat{S}$ of size $n$ as follows. Its first $n_\alpha$ entries are $x^{(1)}, \ldots, x^{(n_\alpha)}$; then for the
90 remaining $n - n_\alpha$, we have (1) the first $\lceil \frac{n - n_\alpha}{2} \rceil$ (i.e. the first half) of those entries are all copies of $\mathbf{c}$,
91 and (2) the remaining $\lfloor \frac{n - n_\alpha}{2} \rfloor$ are copies of $-\mathbf{c}$.

92 Note that we have

$$q(\hat{S}) = \frac{n_\alpha}{n} q(S)$$

93    for every $S$, and in particular $\|q(\hat{S})\|_2$ satisfies the assumption in the "Moreover."

94    Now, we define an algorithm $\hat{\mathcal{A}}$ for approximating $q$ on datasets of size $n_\alpha$ as follows. On input
95    $S \in \{\pm 1\}^{d \times n_\alpha}$, $\hat{\mathcal{A}}$:

96        1. Computes $\hat{S} \in \{\pm 1\}^{d \times n}$ as above

97        2. Outputs $\frac{n}{n_\alpha} \mathcal{A}(\hat{S})$

98    Since $\mathcal{A}$ is already differentially private, $\hat{\mathcal{A}}$ is also $(\varepsilon, \delta)$-DP due to the post-processing property of
99    differential privacy. Moreover,

$$\mathbb{E}\Big[\|\hat{\mathcal{A}}(S) - q(S)\|_2\Big] = \mathbb{E}\bigg[\Big\|\frac{n}{n_\alpha}\mathcal{A}(\hat{S}) - \frac{n}{n_\alpha}q(\hat{S})\Big\|_2\bigg] = \frac{n}{n_\alpha}\mathbb{E}\Big[\Big\|\mathcal{A}(\hat{S}) - q(\hat{S})\Big\|_2\Big] \overset{(5)}{<} \frac{n}{n_\alpha} \cdot \frac{n_\alpha}{n}\alpha = \alpha$$

100    and so $\hat{\mathcal{A}}$ achieves expected error strictly smaller than $\alpha$ on datasets of size $n_\alpha$; which contradicts
101    the first part of the lower bound we already established. So for $n > n_\alpha$, any $(\varepsilon, \delta)$-DP algorithm to
102    estimate $q$ must have expected $L_2$ error at least $\frac{n_\alpha}{n} \cdot \alpha = C \cdot \frac{\sqrt{d \ln(1/\delta)}}{n\varepsilon}$.

103    Finally, we we have shown that for every $n$ and every $\varepsilon > 0$, there is a constant $C > 0$ such that every
104    $(\varepsilon, \delta)$-differentially private algorithm $\mathcal{A}$ answering the linear query $q$ must have, on some dataset $S$ of
105    size $n$, expected $L_2$ error at least

$$\mathbb{E}[\|\mathcal{A}(S) - q(S)\|_2] = \min\left(\alpha, C \cdot \frac{\sqrt{d \ln(1/\delta)}}{n\varepsilon}\right).$$

106    proving the lemma.        $\square$

107    Combining the above with the argument strategy of [Bassily et al., 2014, Theorem 5.3] finally yields
108    the main lemma for the second step of our proof for Theorem 1.1:

109    **Lemma 2.4** (Lower bound on empirical loss of $(\varepsilon, \delta)$-DP algorithms). *Let $n, d \in \mathbb{N}, \varepsilon > 0$, and*
110    *$\delta = o(1/n)$. For every $(\varepsilon, \delta)$-differentially private algorithm with output $\theta^{priv}$, there is a dataset*
111    *$S = \{x_1, \ldots, x_n\} \subseteq \{-\frac{1}{\sqrt{d}}, \frac{1}{\sqrt{d}}\}^d$ such that*

$$\mathbb{E}\big[\mathcal{L}(\theta^{priv}, S) - \mathcal{L}(\theta^*, S)\big] = \min\left(\alpha^2, \Omega\left(\frac{d \log(1/\delta)}{n^2 \varepsilon^2}\right)\right)$$

112    *where $\theta^* := \frac{\sum_{i=1}^n x_i}{\|\sum_{i=1}^n x_i\|_2}$ is the minimizer of $\mathcal{L}(\theta, S) := -\langle \theta, \frac{1}{n}\sum_{i=1}^n x_i \rangle$ (which is linear and, as*
113    *such, Lipschitz and convex).*

114    *Proof of Lemma 2.4.* This proof follows the same structure as that of [Bassily et al., 2014, Theo-
115    rem 5.3] but adapt the bound in terms of expectation.

116    First, observe that for any $\theta \in \mathbb{B}$ and dataset $S$ we have:

$$\mathcal{L}(\theta, S) - \mathcal{L}(\theta^*, S) = \frac{1}{2}\|q(S)\|_2\|\theta - \theta^*\|_2^2,$$

117    since $\|\theta - \theta^*\|_2^2 = \|\theta^*\|_2^2 + \|\theta\|_2^2 - 2\langle \theta, \theta^* \rangle = 2(1 - \langle \theta, \theta^* \rangle)$ using the fact that $\theta^*, \theta \in \mathbb{B}$ have
118    $\|\theta\|_2, \|\theta^*\|_2 = 1$.

119    Suppose that there is an $(\varepsilon, \delta)$-differentially private algorithm $\mathcal{A}$ that outputs $\theta^{priv}$ such that, for
120    every dataset $S \subseteq \{-\frac{1}{\sqrt{d}}, \frac{1}{\sqrt{d}}\}^d$, we have:

$$\mathbb{E}\big[\mathcal{L}(\theta^{priv}, S) - \mathcal{L}(\theta^*, S)\big] \leq \Delta$$

121    for a sufficiently small constant $C > 0$, and some $\Delta \geq 0$. We will prove a lower bound on $\Delta$. To
122    do so, recall $q(S) = \theta^* \cdot \|q(S)\|_2$; and that the lower bound from Lemma 2.3 still holds when the
123    dataset $S$ is promised to be such that $q(S) \in [(M \pm 1)/n]$, for $M = \Theta(\min(n\alpha, \sqrt{d \log(1/\delta)}/\varepsilon))$.

124 Consider the algorithm (private by post-processing) $\mathcal{A}$ which outputs $\mathcal{A}(S) = \frac{M}{n}\theta^{priv}$. Then, for
125 any dataset $S$ such that $\|\sum_{i=1}^{n} x_i\|_2 \in [M-1, M+1]$,

$$\mathbb{E}[\|\mathcal{A}(S) - q(S)\|_2] \leq \mathbb{E}[\|\mathcal{A}(S) - q(S)\|_2^2]^{1/2} = \mathbb{E}\left[\|\frac{M}{n}\theta^{priv} - q(S)\|_2^2\right]^{1/2}.$$

126 On the other hand,

$$\mathbb{E}\left[\|\frac{M}{n}\theta^{priv} - q(S)\|_2^2\right] \leq 2\left(\mathbb{E}\left[\|q(S)\|_2^2\|\theta^{priv} - \theta^*\|_2^2\right] + \mathbb{E}\left[\|\frac{M}{n}\theta^{priv} - \|q(S)\|_2\theta^{priv}\|_2^2\right]\right)$$

$$= 4\|q(S)\|_2\mathbb{E}\left[\mathcal{L}(\theta^{priv}, S) - \mathcal{L}(\theta^*, S)\right] + 2\left(\frac{M}{n} - \|q(S)\|_2\right)^2$$

$$\leq \frac{4(M+1)}{n}\mathbb{E}\left[\mathcal{L}(\theta^{priv}, S) - \mathcal{L}(\theta^*, S)\right] + \frac{2}{n^2}$$
$$\text{(as } n\|q(S)\|_2 \in [M-1, M+1])$$

$$\leq \frac{4(M+1)\Delta}{n} + \frac{2}{n^2}$$

127 By Lemma 2.3, we know that $\mathbb{E}[\|\mathcal{A}(S) - q(S)\|_2] = \min\left(\alpha, C \cdot \frac{\sqrt{d\ln(1/\delta)}}{n\varepsilon}\right)$, for some absolute
128 constant $C > 0$, in the worst case. Hence, we must have

$$\frac{\Delta \cdot M}{n} \geq \min\left(\alpha^2, \frac{d\ln(1/\delta)}{n^2\varepsilon^2}\right);$$

129 recalling the setting of $M$, we get $\mathbb{E}\left[\mathcal{L}(\theta^{priv}, S) - \mathcal{L}(\theta^*, S)\right] = \min\left(\alpha, \Omega\left(\sqrt{\frac{d\ln(1/\delta)}{n\varepsilon}}\right)\right).$ $\square$

130 The above lemma establishes a lower bound on the empirical loss of any $(\varepsilon, \delta)$-differentially private
131 algorithm. To derive from this our claimed lower bound on unlearning algorithms, we need to
132 introduce a dependence on $m$, the deletion capacity (i.e., number of points to unlearn). This is done
133 in the last (third) step of our argument, via a reduction which establishes a (restricted) converse to the
134 groupposition property of DP.

135 Recall that an algorithm $M: \mathcal{X}^n \to \mathcal{Y}$ satisfies $(\varepsilon, \delta)$-DP for edit distance $m$ if for every pair of
136 neighboring datasets $X, X'$ *that differ in up to $m$ entries*, and every $S \subseteq \mathcal{Y}$:

$$\Pr[M(X) \in S] \leq e^{\varepsilon}\Pr[M(X') \in S] + \delta.$$

137 We apply this $m$-edit distance $(\varepsilon, \delta)$-DP on Lemma 2.4 by a reduction that shows: for any differentially
138 private algorithm with respect to edit distance at most $m$ must incur an empirical loss given by
139 Lemma 2.4.

140 **Lemma 2.5.** *Suppose there exists an $m$-edit distance $(\varepsilon, \delta)$-DP algorithm $\mathcal{M}$ that takes in a dataset*
141 *$S$ of size $n$ to approximate $q(S)$ (as defined in (3)), with empirical loss $\gamma$. Then, we can construct a*
142 *1-edit distance (i.e., standard) $(\varepsilon, \delta)$-DP algorithm $\mathcal{M}'$ that, on input a dataset $S'$ of $N = n/m$ data*
143 *points, approximates $q(S')$ to error $\gamma$.*

144 *Proof of Lemma 2.5.* The reduction is quite simple: given $\mathcal{M}$, construct $\mathcal{M}'$ as follows for $N = \frac{n}{m}$
145 inputs:

$$\mathcal{M}'(x_1, \ldots, x_N) = \mathcal{M}(\underbrace{x_1, \ldots, x_1}_{m}, \underbrace{x_2, \ldots, x_2}_{m}, \ldots, \underbrace{x_N, \ldots, x_N}_{m}).$$

146 We immediately have that $\mathcal{M}'$ is $(\varepsilon, \delta)$-DP for the usual 1-edit distance between datasets, since
147 $\mathcal{M}$ is DP with respect to edit distance $m$. The sample complexity and error bound then follows
148 direction from $n = N \times m$, where $n \geq N, N \in \mathbb{N}, m \geq 1$, and the fact that $q(x_1, \ldots, x_N) =$
149 $q(x_1, \ldots, x_1, x_2, \ldots, x_2, \ldots, x_N, \ldots, x_N)$ due to the definition of $q$. $\square$

150 Combining Lemma 2.5 with Lemma 2.4, we get that any $m$-edit distance $(\varepsilon, \delta)$-DP algorithm $\mathcal{M}$
151 approximating $q$ on datasets of size $n = N \times m$ must have error $\gamma$ at least

$$\min\left(\alpha, \Omega\left(\frac{\sqrt{d\log(1/\delta)}}{N\varepsilon}\right)\right) = \min\left(\alpha, \Omega\left(\frac{m\sqrt{d\log(1/\delta)}}{n\varepsilon}\right)\right)$$

152 which, reorganising the terms and recalling the definition of deletion capacity, yields the claimed
153 bound on $m_{\varepsilon,\delta}^{A,\bar{A}}$, and hence completes the proof for Theorem 2.1. □

154 The proof of Theorem 1.2 (the strongly convex case), restated below, is analogous to those of
155 Theorems 1.1 and 2.1, but using [Feldman et al., 2020, Theorem 5.1] for the upper bound (in-
156 stead of Lemma 1.3) and [Steinke and Ullman, 2016, Theorem 5.2] for the lower bound (instead
157 of Theorem 2.2).

158 **Theorem 2.6** (Unlearning For Strongly Convex Loss Functions (Theorem 1.2, restated)). *Let $f: \mathcal{W} \times$*
159 *$\mathcal{X} \to \mathbb{R}$ be a 1-Lipschitz strongly convex loss function. There exists an $(\varepsilon, \delta)$-machine unlearning*
160 *algorithm which, trained on a dataset $S \subseteq \mathcal{X}^n$, does not store any side information about the training*
161 *set besides the learned model, and can unlearn up to*

$$m = O\left(\frac{n\varepsilon\sqrt{\alpha}}{\sqrt{d\log(1/\delta)}}\right)$$

162 *datapoints without incurring excess population risk greater than $\alpha$. Moreover, this is tight.*

# 3 Proof of $(\varepsilon, \delta)$-unlearning properties

164 The laziness assumption defined below is essential for the proof, and a natural requirement for
165 practical applications.

166 **Assumption 3.1** (Unlearning Laziness (Assumption 1.3 in Submission)). *An unlearning algorithm*
167 *$(\bar{A}, A)$ is said to be* lazy *if, when provided with an* empty *set of deletion requests, the unlearning*
168 *algorithm $\bar{A}$ does not update the model. That is, $\bar{A}(\emptyset, A(X), T(X)) = A(X)$ for all datasets $X$.*

169 **Theorem 3.2** (Post-processing of unlearning (Theorem 1.4 in Submission)). *Let $(\bar{A}, A)$ be an*
170 *$(\varepsilon, \delta)$-unlearning algorithm taking no side information. Let $f: \mathcal{W} \to \mathcal{W}$ be an arbitrary (possibly*
171 *randomized) function. Then $(f \circ \bar{A}, A)$ is also an $(\varepsilon, \delta)$-unlearning algorithm.*

172 *Proof.* The proof follows exactly same as post-processing property of differential privacy. We
173 consider the case that $f$ is a deterministic function here without loss of generality.

174 Let $T = \{r \in \mathbb{R}^d \mid f(r) \in \mathcal{Y}\}$ and $\mathcal{Y} \subseteq \mathbb{R}^d$. Consider for any $\mathcal{Y} \subseteq \mathbb{R}^d$:

$$\begin{aligned}
\Pr\big[\, f(\bar{A}(A(S), U)) \in \mathcal{Y} \,\big] &= \Pr\big[\, \bar{A}(A(S), U) \in T \,\big] \\
&\leq e^\varepsilon \Pr\big[\, \bar{A}(A(S), U) \in T \,\big] + \delta \\
&= e^\varepsilon \Pr\big[\, f(\bar{A}(A(S), U)) \in \mathcal{Y} \,\big] + \delta
\end{aligned}$$

175 □

176 Under our laziness assumption, we can establish bounds on applying unlearning algorithm repeatedly
177 when the overall deletion requests is within the deletion capacity:

178 **Theorem 3.3** (Chaining of unlearning (Theorem 1.5 in Submission)). *Let $(\bar{A}, A)$ be a lazy $(\varepsilon, \delta)$-*
179 *unlearning algorithm taking no side information, and able to handle up to $m$ deletion requests. Then,*
180 *the algorithm which uses $(\bar{A}, A)$ to sequentially unlearn $k$ disjoint deletion requests $U_1, \ldots, U_k \subseteq X$*
181 *such that $|\cup_i U_i| \leq m$, outputting*

$$\bar{A}(U_k, \ldots, \bar{A}(U_1, A(X))\ldots)$$

182 *is an $(\varepsilon', \delta')$-unlearning algorithm, with $\varepsilon' = k\varepsilon$ and $\delta' = \delta \cdot \frac{e^{k\varepsilon}-1}{e^\varepsilon-1} = O(k\delta)$ (for $k = O(1/\varepsilon)$).*

183 *Proof.* We proceed by induction on $n \geq 1$. Given a pair of $(\varepsilon, \delta)$-unlearning algorithm $(\bar{A}, A)$ and
184 deletion requests $D_1, \ldots, D_n \subseteq S \in \mathbb{R}^{n \times d}$ such that $|\cup_i D_i| \leq m_{\varepsilon,\delta}^{\bar{A},A}$ with $D_i \cap D_j =, \forall i \neq j$ for
185 $i, j \in [n]$.

186 (1) For $n = 1$:
$$\Pr\big[\, \bar{A}(A(S), D_1) \in T \,\big] \leq e^{n\varepsilon} \Pr\big[\, \bar{A}(A(S \setminus D_1), \emptyset) \,\big] + \delta$$

187 by the definition of $(\varepsilon, \delta)$-unlearning. Hence the case $n = 1$ holds.

188 (2) Assume $n = k$ is true:

$$\Pr\big[\,\bar{A}(\ldots\bar{A}(A(S), D_1), \ldots, D_k) \in T\,\big] \leq e^{k\varepsilon} \Pr\big[\,\bar{A}(A(S \setminus \bar{D}_k), \emptyset)\,\big] + \sum_{i=0}^{k-1} e^{i\varepsilon} \cdot \delta \qquad (6)$$

189 (3) Then for $n = k + 1$:

$$\Pr\big[\,\bar{A}(\ldots\bar{A}(A(S), D_1), \ldots, D_{k+1}) \in T\,\big] \overset{(6)}{\leq} e^{k\varepsilon} \Pr\big[\,\bar{A}(\bar{A}(A(S \setminus \bar{D}_k), \emptyset), D_{k+1})\,\big] + \sum_{i=0}^{k-1} e^{i\varepsilon} \cdot \delta$$

$$= e^{k\varepsilon} \Pr\big[\,\bar{A}(A(S \setminus \bar{D}_k), D_{k+1})\,\big] + \sum_{i=0}^{k-1} e^{i\varepsilon} \cdot \delta$$

$$\leq e^{(k+1)\varepsilon} \Pr\big[\,\bar{A}(A(S \setminus \bar{D}_{k+1}), \emptyset) \in T\,\big] + \sum_{i=0}^{(k+1)-1} e^{i\varepsilon} \cdot \delta$$

190 where the first and third inequality result from the definition of $(\varepsilon, \delta)$-unlearning and the second
191 equality is due to Laziness Assumption 3.1.

192 Hence, by induction, the claim holds for all $n \in \mathbb{N}$. $\qquad\square$

193 **Theorem 3.4** (Advanced composition of unlearning (Theorem 1.6 in Submission))**.** *Let*
194 $(\bar{A}_1, A), \ldots, (\bar{A}_k, A)$ *be* lazy $(\varepsilon, \delta)$-unlearning (with common learning algorithm A) taking no
195 *side information, and define the composition of those unlearning algorithms, $\tilde{A}$ as*

$$\tilde{A}(U, A(X)) = f\big(\bar{A}_1(U, A(X)), \ldots, \bar{A}_k(U, A(X))\big).$$

196 *where* $f: \mathcal{W}^k \to \mathcal{W}$ *is any (possibly randomized) function. Then, for every $\delta' > 0$, $(\tilde{A}, A)$ is an*
197 $(\varepsilon', \delta')$-*unlearning taking no side information, where* $\varepsilon' = \frac{k}{2}\varepsilon^2 + \varepsilon\sqrt{2k \ln (1/\delta')}$.

198 *Proof.* The proof follows the same argument as in [Vadhan, 2017, Lemma 2.4]. We consider the case
199 of $\delta > 0$ only as the $\delta = 0$ is same with the pure DP proof.

200 Fix two datasets, $S$ (original dataset) and $S' := S \setminus U$ ("forgotten dataset") where $U$ is the set of
201 delete requests with $|U| \leq m_{\varepsilon,\delta}^{\bar{A},A}$. Note that $S, S'$ differs in $m$ entries.

202 For an output $y = (y_1, \ldots, y_k) \in \mathcal{Y}$, define "memory" loss (which is just privacy loss in differential
203 privacy) to be:

$$\mathcal{L}_{\mathcal{A}}^{S \to S'}(y) = \ln \frac{\Pr[\,\mathcal{A}(A(S), U) = y\,]}{\Pr[\,\mathcal{A}(A(S'), \emptyset) = y\,]}$$

204 where $|\mathcal{L}_{\mathcal{A}}^{S \to S'}(y)| \leq \varepsilon$.

205 Then, by [Vadhan, 2017, Lemma 1.5] we know that $\bar{A}_i(A(S), U), \bar{A}_i(A(S'), \emptyset)$ are $(\varepsilon, \delta)$-
206 indistinguishable, hence there are events $E = E_1 \wedge \ldots \wedge E_k, E' = E'_1 \wedge \ldots \wedge E'_k$ such that
207 w.p. at least $1 - k\delta$ by, for all $y_i, i \in [k]$,

$$\mathbb{E}\Big[\mathcal{L}_{\mathcal{A}}^{S \to S'}(y)\Big] = \mathbb{E}\Big[\ln \frac{\Pr[\,\mathcal{A}(A(S), U) = y \mid E\,]}{\Pr[\,\mathcal{A}(A(S'), \emptyset) = y \mid E'\,]}\Big]$$

$$= \sum_{i=1}^{k} \mathbb{E}\Big[\ln \Big(\frac{\Pr\big[\,\bar{A}_i(A(S), U) = y \mid E_i\,\big]}{\Pr\big[\,\bar{A}_i(A(S'), \emptyset) = y \mid E'_i\,\big]}\Big)\Big]$$

$$= \sum_{i=1}^{k} \mathbb{E}\Big[\mathcal{L}_{\bar{A}_i}^{S \to S'}(y)\Big]$$

208 where we observe that the expectation of the loss is just KL-divergence between the distributions of
209 $\bar{A}_i(A(S), U)$ and $\bar{A}_i(A(S'), \emptyset)$ conditioned on $E$ and $E'$. Hence:

$$\mathbb{E}\Big[\mathcal{L}_{\mathcal{A}}^{S \to S'}(y)\Big] = \sum_{i=1}^{k} D_{\mathrm{KL}}(\bar{A}_i(A(S), U) \| \bar{A}_i(A(S'), \emptyset)) \leq \frac{k}{2}\varepsilon^2$$

where the inequality is a result from [Bun and Steinke, 2016, Proposition 3.3] when $\alpha = 1$. This proposition is applicable because the conditional distribution of $\bar{A}_i$ is $(\varepsilon, \delta)$-indistinguishable, which shares the max-divergence definition.

Then by Hoeffding's inequality where the loss is bounded by $[-\varepsilon, \varepsilon]$, with probability at least $1 - \delta'$, we have:

$$\exp\left(-\frac{t^2}{2k\varepsilon^2}\right) \geq \Pr\left[\mathcal{L}_{\mathcal{A}}^{S \to S'}(y) > \mathbb{E}\left[\mathcal{L}_{\mathcal{A}}^{S \to S'}(y)\right] + t\right]$$

$$\geq \Pr\left[\mathcal{L}_{\mathcal{A}}^{S \to S'}(y) > \frac{k}{2}\varepsilon^2 + t\right]$$

$$= \Pr\left[\mathcal{L}_{\mathcal{A}}^{S \to S'}(y) > \varepsilon'\right]$$

Now for $\delta' := \exp\left(-\frac{t^2}{2k\varepsilon^2}\right)$, we have $t = \varepsilon\sqrt{2k\ln(1/\delta')}$ and $\varepsilon' := \frac{k}{2}\varepsilon^2 + \varepsilon\sqrt{2k\ln(1/\delta')}$.

Hence, for any set $T \in \mathcal{Y}$:

$$\Pr[\mathcal{A}(A(S), U) \in T] \leq \Pr\left[\mathcal{L}_{\mathcal{A}}^{S \to S'}(y) > \varepsilon'\right] + \sum_{y \in T : \mathcal{L}_{\mathcal{A}}^{S \to S'}(y) \leq \varepsilon'} \Pr[\mathcal{A}(A(S), U) = y]$$

$$\leq \delta' + \sum_{y \in T : \mathcal{L}_{\mathcal{A}}^{S \to S'}(y) \leq \varepsilon'} e^{\varepsilon'} \Pr[\mathcal{A}(A(S'), \emptyset) = y]$$

$$\leq \delta' + e^{\varepsilon'} \Pr[\mathcal{A}(A(S'), \emptyset) \in T]$$

where the second inequality is from the definition of unlearning. Thus, along with an application of [Vadhan, 2017, Lemma 1.5], this proves that $\mathcal{A} = (\bar{A}_1, \ldots, \bar{A}_k)$ is indeed $(\varepsilon', \delta' + k\delta)$-unlearning w.r.t. learning algorithm $A$. $\qquad\square$