# OpenReview forum: "Tight Bounds for Machine Unlearning via Differential Privacy"
_NeurIPS.cc/2023/Conference — Submitted to NeurIPS 2023_

### Official Review · Reviewer_Zwzy · 2023-07-04

**Soundness:** 3 good
**Presentation:** 2 fair
**Contribution:** 2 fair
**Rating:** 4
**Confidence:** 3

**Summary:**

This paper studies the machine unlearning problem from the perspective of differential privacy. Specifically, the authors propose to use differentially private models directly so that unlearning update is not necessary (or unlearning is an identity map), and the motivation is to make the unlearning procedure independent of side information (i.e., original training set) to avoid privacy leakage. A tight lower bound on the number of data points that such kind of DP model can be unlearned is shown in the paper, along with some new proving techniques.

**Strengths:**

This paper uses the idea from Renyi DP and zero-concentrated DP to optimize the DP parameters, and thus refine the lower bound result proposed in previous work by Sekhari et al. An interesting combination of off-the-shelf results from DP literature is used to obtain the improvement of deletion capacity lower bound from $\widetilde{\Omega}\left(\frac{n \varepsilon}{\sqrt{d \log \left(e^\varepsilon/ \delta\right)}}\right)$ to $\Omega\left(\frac{n \varepsilon}{\sqrt{d \log (1 / \delta)}}\right)$. Furthermore, a deletion capacity upper bound is studied when the loss function is linear, showing that the proposed lower bound is actually tight.

Besides the improvement in the lower bound, the authors also introduce the post-processing, chain rule, and composition theorem for unlearning analog to classical DP. This could benefit future works to study unlearning via differential privacy.

**Weaknesses:**

Although this paper may be interesting in theory, I do not think it can fit nicely into machine unlearning problems. The main motivation in the paper to use DP directly for unlearning is to avoid the usage of side information, which basically means the original training set. However, for most unlearning scenarios, having the entire training set is not a critical issue, and sometimes even a must when we need to perform model retraining. Consider the case when the size of data points that require deleting is larger than the deletion capacity. We have no choice but to retrain the model from scratch, and we need the remaining training samples to complete the retraining. So I do not think the motivation holds in the first place. Furthermore, a related discussion on the difference between DP and unlearning appears in the previous work by Sekhari et al. (Section 3.2: Strict separation between unlearning and differential privacy), where they show that if one designs the unlearning update carefully, the deletion capacity can be improved to $\Omega(\frac{n \sqrt{\varepsilon}}{(d \log (1 / \delta))^{1 / 4}})$, which enjoys better dependence on the dimension $d$ even compared to the results presented in this paper. As a matter of fact, many unlearning papers have pointed out the fact that using DP directly can lead to large overhead and low utility. Therefore, whether one can use the theoretical results presented in the paper to design better unlearning algorithms is questionable.

**Questions:**

Please see the weakness section for more details.

**Limitations:**

The results do not fit nicely with machine unlearning problems, and it would be hard to utilize the results to design unlearning algorithms.

---

> ### Author Rebuttal · Authors · 2023-08-09
>
> We appreciate the reviewer’s reading of our paper, and thank them for their comments; however, we believe that their assessment of our work hinges on a few misunderstandings, that we hope to clarify below:
> * Regarding the situation where the number of deletion requests exceeds the deletion capacity. Indeed, in this case, retraining from scratch on the remaining samples (or on an entirely new dataset, which is often unrealistic) is necessary. Which actually makes the case for understanding what the deletion capacity exactly is (to know when this threshold is reached after many typically small unlearning requests), which our paper does. In addition, this would typically happen after many requests: meaning that one can design (consistent with the motivation of our paper) unlearning settings where the original dataset is kept in a secure (albeit hard to access) location, and only needs to be accessed very rarely – namely, when this deletion capacity has been reached. This mitigates data breach risks, while still allowing unlearning in the setting considered.
> * The result of Sekhari et al. mentioned by the reviewer does achieve better deletion capacity by storing additional information (namely, an additional statistic $T(S)$ of roughly $d^2$ bits about the dataset $S$), so while the deletion capacity is better this does not address the main concern considered in our work: offering **both** privacy and unlearning.
> * Regarding the overhead due to DP algorithms. Indeed, DP algorithms typically incur a cost in terms of utility; we refer the reviewer to our response to Reviewer tB6F for a discussion of our motivation in that regard. In short, we do not deny the utility cost of DP, and as a result are not advocating using DP *when not required* just to achieve unlearning; instead, our motivation/use case is when privacy **is** required **and** unlearning is requested, in which case combining the two can (as our paper shows) come as essentially no cost until a large number of deletion requests is made (the deletion capacity, which our work pinpoints for convex and strongly convex losses).
>
> We hope that our response clarifies some of the points raised by the reviewer, and will lead them to increase their score.

---

> > ### Comment · Reviewer_Zwzy · 2023-08-13
> > **Reviewer Response**
> >
> > I would like to thank the authors for making efforts to answer the questions from us. After going through all the comments, I feel like the major concern of most reviewers is how the proposed results can benefit the unlearning community. To elaborate, the authors show that when we are allowed to store some additional statistics $T(S)$ about the original dataset, we can get better results on deletion capacity; the proposed result can only outperform previous ones when we are not allowed to use any side information. In other words, if I understand correctly, the authors try to claim that the proposed result respects privacy more strictly, while previous results are only a trade-off between privacy and model utility. However, the "trade-offs" here are hard to define, as DP itself is also another trade-off via the $(\epsilon, \delta)$ parameters. So for applications like unlearning where we do have access to the original data, why we need to exclude additional information beyond the model needs to be justified more clearly. This is also one of the reasons I believe why most unlearning works will include some kind of experiments to show the final performance as a way to justify their method.
> >
> > Personally, I really would like to see more theoretical papers in the field of unlearning. However, for such endeavors to be impactful, there needs to be a solid foundation of well-articulated motivations and assumptions. Given the current scope and clarity, I would like to keep the score at this moment.

---

> > > ### Author Response · Authors · 2023-08-16
> > >
> > > We thank the reviewer for their comments. We do agree that empirical and practical assessment of any proposed solution would be important (and necessary) before deployment. However, in the case of our work, we are not quite clear on what experiments would be meaningful and useful, given that our results either rely on analyzing theoretically the use of existing algorithms (DP) for unlearning, or on lower bounds (which as such are not amenable to experiments). What type of experiments do you have in mind?

---

> > > > ### Comment · Reviewer_Zwzy · 2023-08-21
> > > >
> > > > Sorry for the misunderstanding, but I am not trying to say that no simulation is a drawback of this paper. I totally understand that this is a theory paper and it would be hard to verify the lower bound empirically. I am just suggesting the authors think about how to better explain the benefits of the proposed theory to both DP (theory) and unlearning (practice) community.

---

> > > > > ### Author Response · Authors · 2023-08-22
> > > > >
> > > > > Thank you for clarifying!

---

### Official Review · Reviewer_Hbaj · 2023-07-05

**Soundness:** 3 good
**Presentation:** 3 good
**Contribution:** 2 fair
**Rating:** 4
**Confidence:** 3

**Summary:**

This paper studies connections between machine unlearning and differential privacy (DP). In machine unlearning, the goal is to remove up to, say, $m$ of the examples from a dataset of size $n$, in such a way that the produced model is close (in some form of statistical or computational distance) to the model that would have been produced if we did not have the m examples to begin with.

A recent paper of SAKS'21 formulates unlearning with the same style that DP is usually defined. This paper follows the same definition.

The main question studied in this paper is about "deletion capacity". Namely, how many examples from the dataset can we delete while we lose the accuracy of the model up to a given parameter $\alpha$ and keep the machine unlearning "secure" with specified parameters $(\epsilon,\delta)$ (defined similarly to DP)? More formally, $\alpha$ is the regret in the agnostic setting, which is the extra risk compared to the best model in the family.

The main result of the paper is to establish matching upper and lower bounds (up to constant factors) on the deletion capacity for algorithms that basically do not do any deletion and when certain (natural, but still limiting) properties hold on the models and the loss function. Namely, the paper studies how to achieve $(\epsilon,\delta)$ privacy when the comparison is made between datasets that have hamming distance $m$, rather than $1$, and while the regret is bounded by $\alpha$.

At a technical level, the paper achieves its tight upper and lower bound (on the deletion capacity, within its own defined framework) by actually *not* caring about achieving differential privacy in its standard sense and directly aiming to satisfy the DP Lipschitz property over $m$-close databases out of the box. To achieve tight bounds the paper moves to other notions of DP (based on Reny divergence) first and then comes back to DP after a more effective composition theorem (that exist for such DP style definitions) is applied.

The paper also studies composition of unlearning, but only the statements are stated in the paper and no discussion is presented.

**Strengths:**

At a technical level, studying the DP for $m$-close databases (with the motivation of doing nothing for unlearning!) is interesting, and finding tight bounds for such problem is cool. but I would have preferred a more direct depiction of the result up front by saying that what happens here is not really unlearning and is about DP for $m$-close datasets. Then, maybe depicting unlearning as a potential application would be good, since the application to unlearning comes with some limitations that prevent us from using the full capacity of what unlearning allows.

The proofs also look interesting and as far as I could tell, the difference between this work and previous work is explained well.


**Weaknesses:**

As explained above, I think the connection to unlearning is a bit far fetched, as it comes with strong limitations.

(has a related question)
What I understand is that the paper studies tight bounds for settings in which the unlearning is *not even done* and the closeness of the produced models holds due to the DP-like property over $m$-close data sets. I am not sure why this is equivalent to "not storing anything besides the model itself". If they are equivalent, this needs a proof.

Citations are not in good shape. Examples:
The work of Cohen et al is cited for initiating a formal study of "the right to be forgotten", while works like: https://eprint.iacr.org/2020/254 are done earlier.

The main question of the paper is about privacy vs unlearning, which is also studied in previous (uncited) works like
https://www.usenix.org/conference/usenixsecurity20/presentation/salem
https://arxiv.org/abs/2005.02205
https://arxiv.org/abs/2202.03460:


**Questions:**

See the question in the section above. In addition:

why is theorem 1.6 is called a composition theorem? it seems it would be more natural for a composition theorem to allow up to k batches of deletion in an adaptive way.

line 207.5: why is the distribution p and not D ?

line 229: why is F* defined like this, and not like how it was defined in line 209 ?

the def of loss in line 277: why is the loss defined like that? The loss should take a model and a full *labeled* example and then output something. Your notion ignores the label (maybe x itself has the label already?) and is defined for a set (it would be a risk, in that case, and it typically takes average).

**Limitations:**

The paper is clear in that they only study specific "unearning" methods that come with limitations. It is mentioned that the limitation is not to store anything other than the model, but my understanding is that their limitation is to not do anything when unlearning. These seem different to me, but hopefully the author(s) will clarify this limitation in rebuttal.

---

> ### Author Rebuttal · Authors · 2023-08-09
>
> We thank the reviewer for their comments, and are grateful for their pointers to the literature. We agree that our literature review was lacking, and apologize for that: we will make sure to update it (using these pointers, and others) in the final version, to provide a clearer and more accurate picture of the work in this area.
>
> Regarding the technical comments and typos: we will fix the latter (thank you!); as for the comment on the loss function, we look at the general formulation where the vector (data point) includes all information, including label/value; and the quantity defined here is the empirical loss on a dataset – the normalization by n was not included here for convenience, indeed, but would just be a normalizing factor. We will clarify this to avoid any ambiguity.
>
> Finally, regarding the limitations of our work, and in particular of our notion of unlearning as “not even done”, we refer the reviewer to our response to Reviewer tB6F, which hopefully will clarify this aspect.

---

> > ### Comment · Reviewer_Hbaj · 2023-08-14
> > **Ack**
> >
> > thanks for the response.

---

### Official Review · Reviewer_QTS7 · 2023-07-08

**Soundness:** 2 fair
**Presentation:** 3 good
**Contribution:** 2 fair
**Rating:** 5
**Confidence:** 3

**Summary:**

This paper addresses the concept of "machine unlearning" within the framework of differential privacy. The authors provide tight bounds on the maximum number of data points that can be successfully unlearned without significantly impacting the model's accuracy. They also establish the analog of key properties of DP for machine unlearning. The paper introduces novel results for convex and strongly convex loss functions, as well as properties of post-processing and composition of unlearning algorithms.

**Strengths:**

1. The paper addresses the important and practical problem of machine unlearning, which enables individuals to request the removal of their data from trained models.
2. The paper builds upon previous work and provides enhanced theoretical results. It closes the gap between upper and lower bounds on the deletion capacity achievable by differentially private machine unlearning algorithms.

**Weaknesses:**

1. The paper focuses primarily on theoretical analysis and proofs, but it lacks empirical experiments to validate the proposed machine unlearning algorithms.
2. The paper considers convex loss and strongly convex loss functions in its theoretical analysis. While these assumptions may hold for some models and applications, they may not be applicable to a wide range of real-world machine learning models, such as deep learning models, which often involve non-convex loss functions. This limitation restricts the generalizability and practical applicability of the proposed algorithms.

**Questions:**

1. Considering that many real-world machine learning models, such as deep learning models, involve non-convex loss functions, how applicable are the proposed machine unlearning algorithms to these models?
2. How does the deletion capacity impact the overall utility and performance of the machine learning models in practical scenarios?

**Limitations:**

1. Lack of experiments.
2. Assumptions on loss functions are constraints.

---

> ### Author Rebuttal · Authors · 2023-08-09
>
> We thank the reviewer for their time and comments, and address both of their questions together. Our paper does focus on convex and strongly convex losses, as is common in a significant part of the learning and optimization literature; we note that while this assumption on the loss is not always satisfied (as the reviewer correctly points out), analyzing these cases however is a good (and necessary) starting point, and one that provides not only a good rule of thumb but also surprisingly good results overall (as shown by many algorithms, starting with SGD, whose guarantees for convex losses appear to carry over surprisingly well in practice.
>
> Now, as our results do study how DP guarantees imply unlearning ones, the applicability of these algorithms to practical unlearning scenarios then will follow from that of the corresponding DP algorithms for these scenarios. Put differently: design a good and practical differentially private algorithm (as many DP practitioners are working and focusing on), get a good and practical unlearning guarantee from it.
>
> Regarding the lack of experiments: our paper focuses on understanding the interplay between DP and unlearning, and the analogies between the two, from a fundamental point of view. Because of this, we believe that our results stand by themselves (in particular, lower bounds are not amenable to experiments) and are in scope for NeurIPS. We agree that continuing this direction of research further will lead to real-world use down the line, which will require experimental results and evaluation: and we do hope our work will spark interest in this line of research.

---

> > ### Comment · Reviewer_QTS7 · 2023-08-21
> >
> > Thank you for the clarifications. No further questions as of now.

---

### Official Review · Reviewer_RqTt · 2023-07-11

**Soundness:** 3 good
**Presentation:** 3 good
**Contribution:** 2 fair
**Rating:** 5
**Confidence:** 4

**Summary:**

The paper studies approximate unlearning with procedures which do not store any side information (and satisfy differential privacy) in convex learning problems and establishes tight upper and lower bounds.

**Strengths:**

1. Machine unlearning has recently gained much interest owing to privacy regulations. The paper studies a certain formulation of approximate unlearning inspired via differential privacy, with the additional restriction of storing no side information. This formulation is particularly appealing since (a). storing side information can be expensive and prone to attacks and (b). it does not require new algorithms for learning and unlearning, but simply (existing) private algorithms. The paper studies the limits of such a formulation for convex learning problems, improving the upper and lower bounds in the previous works. The question is very natural and the authors resolve the loose ends remaining in the prior works.

2. The proofs of improved upper and lower bound are simple yet interesting: the improved upper bound follows due to the use of stronger group-privacy properties of zCDP, as opposed to approximate DP; and the lower bound establishes an interesting "converse to group-privacy" due the linear loss function under consideration.

**Weaknesses:**

1. While the problem is very natural, the final quantitative improvements are rather minor. For constant $\epsilon$, which is usually the case of DP, the bounds are same. There is also very limited discussion on the quantitative improvements compared to prior work.

2. The scope of the paper seems limited; the paper essentially ties some loose ends present in the prior analysis, which is mostly interesting for theoretical reasons. I don't know if the contributions could have impact on the larger area of machine unlearning.

3. The techniques largely borrow from the differential privacy literature. The lower bound instance and the argument of padding the sample multiple times are present in prior works.

**Questions:**

I would encourage the authors to provide an extended discussion of the quantitative improvements compared to prior work.

**Limitations:**

The scope of the the work is limited to procedures which store no side information and satisfy approximate unlearning (in the spirit of DP). These restrictions basically leave DP procedures as candidate learning/unlearning algorithms.

---

> ### Author Rebuttal · Authors · 2023-08-09
>
> We are grateful to the reviewer for their time and valuable feedback; we address below their main question, regarding quantitative improvement upon previous work; and will incorporate these into the final version of our paper.
>
> Our improvements regarding the deletion capacity of unlearning algorithms (setting aside, for the sake of this discussion, the other contributions of our paper: namely, the extension to “pure” unlearning algorithms, as well as the various additional results regarding, e.g., composition of unlearning algorithms) are twofold:
> 1. The first, the upper bound, can indeed appear minor, in that (as the reviewer points out) for small epsilon it is relatively small. Yet, it is important to mention that while “small epsilon” is usually desired, and the ideal setting, in practice epsilon is typically not small: see, for instance, https://desfontain.es/privacy/real-world-differential-privacy.html for a summary of typical settings in deployments at scale. The value of epsilon is almost always greater than 1, and often set to 8 to 16 (and sometimes even larger). For such values, or improvement becomes non-negligible. (Moreover, this is addressing the practical aspects of the upper bound improvement: we feel important to mention that another key aspect lies in understanding the fundamental limits of this approach (unlearning via DP), from a theoretical and conceptual point of view.)
> 2. The second, the lower bound, goes much beyond this, as the previous lower bound did not feature any meaningful dependence on the deletion capacity at all! In that sense, our results together not only improve upon previous work, they also show what to expect overall – prior to our result, it was by and large open what the “right” limit of deletion capacity was in the large gap left open. The fact that our lower bound shows that this “right” limit happens to be close to the previously known upper bound (again, with the above large v. small epsilon caveat) while our improved upper bound closes the remaining gap does not imply that the situation was roughly well-understood before!
>
> Finally, regarding the limitation, we point the reviewer to our response to Reviewer tB6F regarding the takeaway of our work; namely, that our work is not meant to rule out non-DP approaches to unlearning, but rather one of its main objectives is to deepen our understanding of whatever unlearning guarantees DP brings “for free”, valuable in situations where a DP solution would be required (and thus whichever unlearning guarantee is provided by it comes as an bonus/incentive).

---

> > ### Comment · Reviewer_RqTt · 2023-08-21
> > **Thanks!**
> >
> > I thank the authors for their detailed response. While I agree that this work solidifies our understanding of unleraning guarantees arising from DP based algorithms,the scope is still (in my opinion) narrow, espically since both upper and lower bounds are based on improvement to results in prior work. Nonetheless, I increase my score to 5.

---

> ### Comment · Senior_Area_Chairs · 2023-08-21
> **final discussions**
>
> Dear Reviewer,
>
> As discussions come to an end soon, this is a polite reminder to engage with the authors in discussion.
> Please note we take note of unresponsive reviewers.
>
> Best regards,
> \
> SAC

---

### Official Review · Reviewer_tB6F · 2023-08-01

**Soundness:** 3 good
**Presentation:** 4 excellent
**Contribution:** 3 good
**Rating:** 7
**Confidence:** 3

**Summary:**

This work derives tight bounds for the deletion capacity of machine unlearning algorithms that are differentially private. These bounds are stated in terms of a deletion capacity formulated as a function of data points that can be removed before the estimation risk (an accuracy measure) becomes too large. The estimation risk (or the excess risk) is assessed over the sampling distribution of i.i.d. data points. The authors derived the results in detail for 1-Lipchitz convex loss functions and presented briefly the parallel result for 1-L strongly convex loss functions.

My assessment, consisting of strengths, weaknesses, and questions, can be found in the sections below.


**Strengths:**

I find this a well-written paper. For a technical topic that is focused on the proof, the paper nevertheless walks the reader through while offering both clarity and insight. The structure of the paper is also quite reasonable, with the contribution overview very helpful.


**Weaknesses:**

Please see my question in the Limitation section. I think the lack of comment for that question in this paper is its biggest weakness.

**Questions:**

My biggest concern has been raised as a question in the Limitation section, because it is more appropriate under that heading. Below are a few questions on the technical side that could benefit from some clarification.

- Your estimation risk, and the definition of deletion capacity, calls for an expectation over the sampling variability of i.i.d. data points following a distribution D.  I would like to understand better the impact that this assumption has, in particular the independence part, on your results. In particular, how would data dependence change the results? This is an important point of difference between your framework and DP which do not take data variability into consideration (except Pufferfish which you do not use).
- Related to this, perhaps you could consider including in the sketch proof of Theorem 3.3 (Line 293) some details concerning the reduction from population to empirical losses. I imagine that the issue of data variability comes up here.
- I am failing to appreciate the significance of requiring the data points to take values in \pm 1/\sqrt{d}, where d is the dimension of the parameter space. If we are discussing an asympototic regime (in which the bounds are situated, since they involve both d and n), what does this “shrinking” ball of data mean?
- A minor point: the word “grouposition” can use some clarification. I think I know what the authors mean but this is not standard vocabulary.

**Limitations:**

Looking at the form of the unlearning algorithm \bar{A} in Theorem 3.1, am I to conclude that the unlearning algorithm that is to achieve the tight bound as you present in this work is in fact an algorithm that literally *does nothing* to the deletion request, i.e. it outputs A(S) just as before? If my understanding is correct, this point calls for a moral discussion: yes the learning-unlearning pair satisfies Definition 2.5, but with the relaxation of alpha, epsilon, and delta, this technical argument will provide a slippery slope for justified inaction. Maybe your paper is not the first to consider this, but one more paper gets published without paying due attention to these ethical questions, the literature grows a bit more oblivious to common sense.

---

> ### Author Rebuttal · Authors · 2023-08-09
>
> We are grateful to the reviewer for their time, comments and positive assessment of our work; we address below their questions, starting with the main one (“Limitation”).
>
> Indeed, the reviewer is correct, in that our paper analyzes unlearning algorithms which do not do anything (at deletion request time). However, this is not quite the case that the algorithm “does nothing” overall: instead, the point here is that the algorithms considered benefit from some “deletion bonus” somehow *for free*, as they were designed to already satisfy the very stringent notion of differential privacy (DP) [and, accordingly, paid the price in utility that comes with it].
>
> Put differently, the aim of this paper is not to promote “justified inaction”, but instead to characterize what unlearning guarantee comes “for free” [and when it stops] if one decides to offer the strong guarantee of differential privacy. That is:
> * “Plain” algorithms offer neither privacy nor unlearning
> * Privacy comes at a cost (which has often been argued to be steep)
> * Unlearning comes at a cost
> * Sometimes, only one of the two is required; sometimes, both are desired. Do the “costs” add up, or does paying the cost for DP offering a headstart in terms of unlearning?
>
> Again, our aim is not to discourage unlearning-only solutions when DP is not required; but instead, by understanding the interplay between DP and unlearning, to show that the joint differential privacy+right to be forgotten requirement is more affordable than it seems. We will add a more detailed discussion of this point to clarify it.
>
> Turning to the more technical questions:
> * Indeed, our notion of risk (and the resulting definition of deletion capacity) are linked to the population risk, and our algorithms as a result assume that the dataset is drawn i.i.d. This is necessary to relate empirical loss to population loss, and is the standard setting in learning. While these notions (risk and deletion capacity) do rely on this standard iid assumption, it is important to note that the definition of $(\varepsilon, \delta)$-unlearning (Definition 2.5)  itself does not: and, in that sense, is indeed analogous to the standard definition of DP (which is indeed assumption-free and non-distributional).
> * We will expand on this reduction between empirical and population loss in the supplemental of the paper. As the reviewer correctly mentions, this correspondence is indeed where the i.i.d. assumption on the dataset comes in.
> * This is a good question! The reparameterization to $\{\pm1/\sqrt{d}\}^d$ is mostly for convenience in the lower bound argument, and prevent any unwanted dependence on the dimension to be unaccounted for (e.g., in the Lipschitz constant of the loss function we consider for the lower bound). More precisely, this “shrinking” makes all datapoints considered unit vectors (with respect to $\ell_2$ norm), which simplifies the argument and makes it “cleaner.”

---

> > ### Comment · Reviewer_tB6F · 2023-08-11
> >
> > I thank the authors for taking the time to respond to my questions.

---

### Author Rebuttal · Authors · 2023-08-09

We thank the reviewers for their careful reading of our submission, and are grateful for their detailed comments and suggestions. We will address their specific comments in the final version of our work, and respond individually to their questions and concerns below.

---

### Decision · Program_Chairs · 2023-09-21

**Decision:**

Reject

**Comment:**

The topic of the paper is undeniably interesting *but* the global message, at the end of the interaction phase, remains that it is underwhelming. I attribute it to the fact that the authors try to use the global message that “solving one problem solves another one “for free””, without trying that much to be convincing beyond this message. In my experience, such a bare free lunch is indeed rare and the authors could have done better in the rebuttal(s).

I see two key criticisms on the paper, around its narrow scope (RqTt) and substantial + substantiated criticisms (Hbaj). I do not really understand why the authors have not taken the time to precisely answer those last criticisms, instead pointing to a review (tb6f) which was undeniably positive about the work. If the authors did not want to target their rebuttals, a simple rationale would have rather dictated to answer the most negative criticism and then point out / reformulate for others.

I suggest to have a close look at several key points which, once addressed, would surely put the paper on a better path towards acceptance. For example, RqTt mentions the narrow scope and some incremental nature in the results (with respect to prior work) as presented. In my opinion, the idea of discarding all dependencies in the loss function in the results (they disappear in the big-Oh / Omega) was not a good idea. The choice of the loss is a fundamental problem: first-order information about the loss gives indication on how to reach statistical consistency for some algorithms, second-order information gives indication on rates of convergence. Making the loss dependent parameters to clearly appear would indicate how unlearning plays in the broader scope of not just privacy, but also training rates and generalization.